EMBO
Molecular Medicine

# Human soluble ACE2 improves the effect of remdesivir in SARS-CoV-2 infection

Vanessa Monteil[1] (iD), Matheus Dyczynski[2,3] (iD), Volker M Lauschke[4] (iD), Hyesoo Kwon[5],
Gerald Wirnsberger[6], Sonia Youhanna[4] (iD), Haibo Zhang[7] (iD), Arthur S Slutsky[7] (iD), Carmen Hurtado del Pozo[8,9,10], Moritz Horn[2,3] (iD), Nuria Montserrat[8,9,10] (iD), Josef M Penninger[11,12,*] (iD) & Ali Mirazimi[1,5,**] (iD)

## Abstract

There is a critical need for safe and effective drugs for COVID-19. Only remdesivir has received authorization for COVID-19 and has been shown to improve outcomes but not decrease mortality. However, the dose of remdesivir is limited by hepatic and kidney toxicity. ACE2 is the critical cell surface receptor for SARS-CoV-2. Here, we investigated additive effect of combination therapy using remdesivir with recombinant soluble ACE2 (high/low dose) on Vero E6 and kidney organoids, targeting two different modalities of SARS-CoV-2 life cycle: cell entry via its receptor ACE2 and intracellular viral RNA replication. This combination treatment markedly improved their therapeutic windows against SARS-CoV-2 in both models. By using single amino-acid resolution screening in haploid ES cells, we report a singular critical pathway required for remdesivir toxicity, namely, Adenylate Kinase 2. The data provided here demonstrate that combining two therapeutic modalities with different targets, common strategy in HIV treatment, exhibit strong additive effects at sub-toxic concentrations. Our data lay the groundwork for the study of combinatorial regimens in future COVID-19 clinical trials.

**Keywords** clinical trial; combination therapy; COVID-19; treatment
**Subject Categories** Microbiology, Virology & Host Pathogen Interaction

## Introduction

In December of 2019, a novel coronavirus (SARS-CoV-2) crossed species barriers to infect humans and was effectively transmitted from person to person, leading to a pneumonia outbreak first reported in Wuhan, China (Jiang *et al*, 2020; Zhou *et al*, 2020). This virus causes coronavirus disease-19 (COVID-19) with influenza like symptoms ranging from mild disease to severe lung failure and multi-organ damage, eventually leading to death, especially in older patients with other co-morbidities. SARS-CoV-2 shares multiple similarities with the original SARS-CoV (Lu *et al*, 2020; Zhu *et al*, 2020). The receptor binding domain (RBD) of SARS-CoV-2 is similar to the SARS-CoV RBD, suggesting a possible common host cell receptor. The SARS-CoV receptor Angiotensin-converting enzyme 2 (ACE2) was indeed rapidly identified to also function as the critical cell surface receptor for SARS-CoV-2 (Walls *et al*, 2020; Wan *et al*, 2020; Wrapp *et al*, 2020).

We were the first to shown that ACE2 counterbalances the effects of Angiotensin (Ang) II *in vivo* and thereby protects the heart, kidney, and, importantly, the lung via its enzymatic RAS activity (Crackower *et al*, 2002; Imai *et al*, 2005). Moreover, we showed that ACE2 is the critical SARS-CoV receptor *in vivo* using ACE2 mutant mouse experiments and that SARS-CoV infections and even purified Spike as well as a minimal Spike domain (RBD) can lead to ACE2 downregulation (Kuba *et al*, 2005), explaining why SARS-CoV, and now SARS-CoV-2 infections cause severe lung failure: ACE2 is the receptor for both viruses and downregulation of ACE2 via virus binding results in loss of RAS tissue homeostasis which then drives disease severity. Moreover, ACE2 expression and its regulation in cardiovascular disease, gender being encoded on the X

---

1 Department of Laboratory Medicine, Unit of Clinical Microbiology, Karolinska Institute, Stockholm, Sweden
2 Acus Laboratories GmbH, Cologne, Germany
3 JLP Health GmbH, Vienna, Austria
4 Department of Physiology and Pharmacology, Karolinska Institute, Stockholm, Sweden
5 National Veterinary Institute, Uppsala, Sweden
6 APEIRON Biologics AG, Vienna, Austria
7 Keenan Research Centre for Biomedical Science at Li Ka Shing Knowledge Institute of St. Michael's Hospital, University of Toronto, Toronto, ON, Canada
8 Pluripotency for Organ Regeneration, Institute for Bioengineering of Catalonia (IBEC), The Barcelona Institute of Technology (BIST), Barcelona, Spain
9 Catalan Institution for Research and Advanced Studies (ICREA), Barcelona, Spain
10 Centro de Investigación Biomédica en Red en Bioingeniería, Biomateriales y Nanomedicina, Madrid, Spain
11 Institute of Molecular Biotechnology of the Austrian Academy of Sciences, Vienna, Austria
12 Department of Medical Genetics, Life Sciences Institute, University of British Columbia, Vancouver, BC, Canada
 *Corresponding author. Tel: +46 703 672 573; E-mail: ali.mirazimi@sva.se
 **Corresponding author. Tel: +1 604 827 4128; E-mail: josef.penninger@ubc.ca

chromosome, or aging can explain and contributes to progression and tissue distribution of COVID-19 as a multi-organ disease (Imai *et al*, 2010). Recently, we reported that human recombinant soluble ACE2 (hrsACE2; APN01) can significantly block early SARS-CoV-2 infections by a factor of 1,000–5,000 (Monteil *et al*, 2020). HrsACE2 entered several clinical trials (Haschke *et al*, 2013; Khan *et al*, 2017). Moreover, we recently described the first named patient treatment with hrsACE2 of a patient with severe COVID-19 (Zoufaly *et al*, 2020). HrsACE2 has entered a placebo controlled, double-blind, phase 2b trial in severe COVID-19 patients (www.c linicaltrials.gov, NCT04335136), acting as a molecular decoy to block virus entry, and as a regulator of the renin–angiotensin system.

As the only drug, remdesivir has received FDA approval for the treatment of COVID-19 (U.S. Food and Drug Administration, 2020). Remdesivir is a RNA polymerase inhibitor, which acts as an adenosine analog, incorporating into nascent viral RNA chains, thus leading to their premature termination (Gordon *et al*, 2020a; Gordon *et al*, 2020b). Initially developed to block replication of Ebola and Marburg viruses, remdesivir shows also anti-viral activity against coronaviruses including MERS, SARS-CoV, and SARS-CoV-2 (Malin *et al*, 2020; Simonis *et al*, 2020). However, its efficacy on viral load and mortality in severe COVID-19 patients is still unclear (Wang *et al*, 2020b). Moreover, it has been reported that remdesivir can exert hepatic and kidney toxicity (Grein *et al*, 2020; Wang *et al*, 2020b). In this study, we report a single amino resolution screen on remdesivir toxicity uncovering a specific mechanism of remdesivir intracellular activation required for cytotoxity. Importantly, combining two different modalities of virus control, blocking entry via hrsACE2 and blocking intracellular viral RNA replication via remdesivir, results in additive effects in SARS-CoV-2-infected cells and human stem cell-derived kidney organoids, reducing the doses of both hrsACE2 and remdesivir to much lower and safer levels.

## Results and Discussion

We hypothesized that combining two different modalities of anti-viral activity using remdesivir (Gordon *et al*, 2020b) and targeting SARS-CoV-2 entry into cells by hrsACE2 (Monteil *et al*, 2020) might show additive effects. We first used Vero E6 cells as a commonly used and robust model of SARS-CoV-2 infection (Monteil *et al*, 2020). In Vero E6 cells, infected with SARS-CoV-2 at a MOI of 20, both remdesivir and hrsACE2 as single agents significantly reduced virus load in a dose-dependent manner (Fig 1A and B), confirming previous data (Monteil *et al*, 2020; Wang *et al*, 2020a). Based on the above dose responses, we determined the IC50 and IC90 values for both remdesivir and hrsACE2 (Table 1). Of note, previous studies have reported different IC50/IC90 values for remdesivir, which is probably due to the different cell types used, different MOI of infections, different SARS-CoV-2 sub-strains, and/or measurements at different time points post-infection (Frediansyah *et al*, 2020; Jeon *et al*, 2020; Pizzorno *et al*, 2020; Wang *et al*, 2020a). Taken together, performing careful side-by-side comparisons in Vero E6 cells, hrsACE2 and remdesivir exhibit comparable efficacy to reduce SARS-CoV-2 infections.

In remdesivir-treated patients, an elevation of enzymes ALT and AST and creatinine have been reported (Mulangu *et al*, 2019;

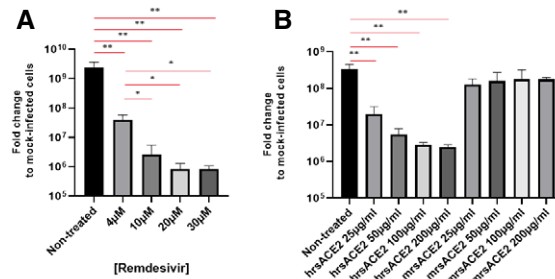

**Figure 1. Blocking entry and replication of SARS-CoV-2 infections.**

A, B (A) Remdesivir and (B) hrsACE2 inhibition of SARS-CoV-2 infections of Vero E6 cells. Both drugs, and murine recombinant soluble ACE2 (mrsACE2, control treatment), were used at the indicated concentrations. Viral RNA level was determined by qRT–PCR 15 h after inoculation of SARS-CoV-2 (Swedish isolate, $10^6$ PFU).

Data information: Error bars show mean $\pm$ SD from biological triplicate. $n = 3$, $*P < 0.05$; $**P < 0.01$; one-way ANOVA followed by Student's *t*-test between internal groups. *P*-values are listed in Appendix Table S1.
Source data are available online for this figure.

Grein *et al*, 2020), indicative of liver and kidney toxicity. To investigate this phenomenon, we have investigated the toxicity of remdisivir in more relevant and advanced in vitro model. Our complex kidney organoids can be readily infected with SARS-CoV-2 (Monteil *et al*, 2020) and are an established model to study various aspects of kidney physiology and pathology (Garreta *et al*, 2019). Liver spheroids of primary human hepatocytes recapitulate the molecular profiles of human liver at the proteomic, transcriptomic, and metabolomics level for multiple weeks in culture (Bell *et al*, 2016; Bell *et al*, 2017; Vorrink *et al*, 2017). Consequently, they outperform other hepatic cell models (Bell *et al*, 2017) and culture paradigms (Bell *et al*, 2018) in multi-center trials and a large toxicity screen using 123 hepatotoxic and non-toxic control drugs found this spheroid system to be the most predictive model for drug-induced liver injury (Vorrink *et al*, 2018). To evaluate cytotoxicity of remdesivir and hrsACE2, we exposed Vero E6 cells, kidney organoids, and liver spheroids to different concentrations of remdesivir alone or in combination with hrsACE2 and assessed cell viability. Remdesivir exhibited significant toxicity in kidney organoids and liver spheroids at doses similar or lower than the effective dose to block SARS-CoV-2 replication (Table 2), in line with clinical findings of liver and kidney injury in patients (Grein *et al*, 2020; Wang *et al*, 2020b). By contrast, CC50 values for hrsACE2 in Vero E6 cells, kidney organoids, and liver spheroids were markedly lower that the effective dose to inhibit the SARS-CoV-2 viral load (Tables 1 and 2). These data show that remdesivir, but not hrsACE2, exhibits liver and kidney toxicity, as determine by engineered human tissues, at doses that are required to effectively control the SARS-CoV-2 infection.

To identify the critical intracellular pathways and interactions that are required for remdesivir cytotoxicity, we performed an unbiased chemical mutagenesis screen in mouse haploid stem cells under strong remdesivir selection (50 μM, Fig 2A). This approach allows to uncover the entire spectrum of mutations resulting in resistance to remdesivir cytotoxicity: loss-of-function, gain-of-function, or neomorph alleles (Horn *et al*, 2018). After ENU

**Table 1.  IC50/IC90 data for remdesivir and hrsACE2 in Vero E6 cells**

| Compound | Cells | MOI | Time post-infection | IC50 (μM) | IC90 (μM) |
|---|---|---|---|---|---|
| Remdesivir | Vero E6 | 20 | 15 h | 4.02 | 5.85 |
| hrsACE2 | Vero E6 | 20 | 15 h | 6.08 | 18.29 |

**Table 2.  CC50 data for remdesivir and hrsACE2 in Vero E6 cells, liver spheroids, and kidney organoids**

| Compound | Vero | Liver | Kidney |
|---|---|---|---|
| hrsACE2 | 6,259 μg/ml | 633 μg/ml | >800 μg/ml |
| Remdesivir | 98.26 μM | 6.77 μM | 10.5 μM |

mutagenesis and remdesivir selection of more than 15 million cells, numerous colonies emerged, whereof the resistance of the five most viable clones was validated (Fig 2B). Whole exome sequencing revealed that all five clones resistant to remdesivir harbor a point mutation in the same unique gene, namely, the *Ak2* gene (Fig 2C). *Ak2* encodes Adenylate kinase 2, an enzyme localized in the mitochondrial intermembrane space that maintains adenine nucleotide homeostasis by catalyzing the reversible reaction AMP + ATP = 2ADP (Noma, 2005). While only one essential isoform is present in bacteria and lower eukaryotes, the genomes of higher vertebrates encode multiple isoforms, suggesting functional redundancy (Liu *et al*, 2019). AK2 is extremely well conserved from mice to humans including conversation of all residues that are mutated in the resistant clones (Fig 2D). One of the four independent AK2 mutations identified resides in a splice acceptor region indicating a loss of function on the protein level. Using molecular modeling, the other three mutations directly or indirectly affect nucleotide binding sites (Fig 2E), pointing to a loss or reduction of AK2 catalytic activity. Given that nucleoside analogs like remdesivir require intracellular enzymatic activation (Eastman *et al*, 2020), our data indicate that remdesivir gets specifically activated by AK2 in the mitochondrial intermembrane space (Fig 2F). Thus, not excluding additional molecules and pathways, our screen has identified a critical enzyme that is required for remdesivir cytotoxicity.

Based on remdesivir toxicity in organoids, we speculated that combining hrsACE2 and remdesivir could enhance their respective anti-viral efficacies and thereby, importantly, reduce the concentrations required for therapeutic effects to doses below the toxic range in our organotypic assays. In healthy volunteers, Cmax of remdesivir was found to be around 7.3 μM after a 225 mg dose (Humeniuk *et al*, 2020), which is in line with loading doses and pharmacokinetics (PK) in critically ill COVID-19 patients (Tempestilli *et al*, 2020). Taking into account these data and our measured IC50 value of remdesivir (Table 1), we used a dose of remdesivir of 4 μM for combination assays. Combinatorial treatment of Vero E6 cells with hrsACE2 (200 μg/ml) and low dose remdesivir (4 μM) indeed reduced the viral load by 60% compared to hrsACE2 alone (Fig 3A). Importantly, we observed similar findings in SARS-CoV-2 infected kidney organoids (Fig 3B). However, at the dose used, hrsACE2 (200 μg/ml) alone

already strongly inhibited viral load and additive effects were not statistically significant. Strikingly, however, low doses of hrsACE2 doses (5 and 10 μg/ml) showed additive effects in combination with low dose remdesivir, resulting in strong and highly significant reduction of SARS-CoV-2 infectivity in Vero E6 cells (Fig 3C) and kidney organoids (Fig 3D). Haschke *et al* have previously reported the pharmacokinetic of hrsACE2, the same molecule we use for our current study, in healthy volunteers in a phase 1 clinical trial (Haschke *et al*, 2013). The human pharmacokinetic data indicate that a concentration of 5–10 μg/ml of hrsACE2 is reached in plasma between 2 and 8 h post-administration by administrating 800 μg/kg. Although one has to await the pharmacokinetic data for hrsACE2 in the phase 2b clinical trial in severe COVID-19 patients, combining hrsACE2 with a viral RNA polymerase inhibitor such as remdesivir should allow to reduce the dose of both drugs and for hrsACE2 to reach an effective *in vivo* anti-viral concentrations, at the doses currently administered to the patients.

Finally, we tested whether this combination treatment would reduce the yield of infectious progeny virus. Intriguingly, treatment with low dose hrsACE2 significantly reduced the yield of progeny virus at 15 hpi to a much larger extent than it inhibited viral RNA (Fig 3E). This effect in the reduction of viral progeny was even more pronounced 48 hpi (Fig 3F). By contrast, low dose remdesivir had only minor effects on the generation of infectious progeny virus at early as well as later time points (Fig 3E and F). Importantly, combining both drugs at low dose resulted in a significantly reduced yield of viral progeny (Fig 3E and F). These data indicate that combination therapies targeting two different critical paths of SARS-CoV-2 infection, namely, viral entry and replication, can enhance the therapeutic effects on viral load and, most importantly, viral progeny.

Previously, we provided the first genetic evidence that ACE2 functions as a negative regulator of the renin–angiotensin system (RAS) in multiple tissues such as the cardiovascular system (Crackower *et al*, 2002). We also showed in genetic experiments that ACE2 is the critical receptor for SARS-CoV *in vivo* and that ACE2 protects the lung from injury, providing a molecular explanation for the severe lung failure and death due to SARS-CoV infections (Imai *et al*, 2005; Kuba *et al*, 2005). The data provided here significantly extend these findings demonstrating that combining two therapeutic modalities with different targets, exhibit strong additive effects at sub-toxic concentrations. The possibility of combining neutralizing antibodies targeting SARS-CoV-2 spike protein with remdesivir has also to be considered and investigated as a therapeutic strategy. These findings are reminiscent of the breakthrough in HIV therapeutics that was bought about by simultaneous targeting of multiple distinct pathways in the viral replication cycle (Gulick *et al*, 1997; Hammer *et al*, 1997). 2D-cellular models (like Vero/Vero E6 cells) are an important piece of data to initiate clinical studies and are a system commonly used for drug screening but most of the drugs tested fail in clinical trial, highlighting the limitation of the system. Using human organoid model for drugs testing might lead to a better selection of drugs that could pass the clinical trial. Even if it does not assure a success in clinical trial, our data lay the groundwork for the study of combinatorial regimens in future COVID-19 clinical trials.

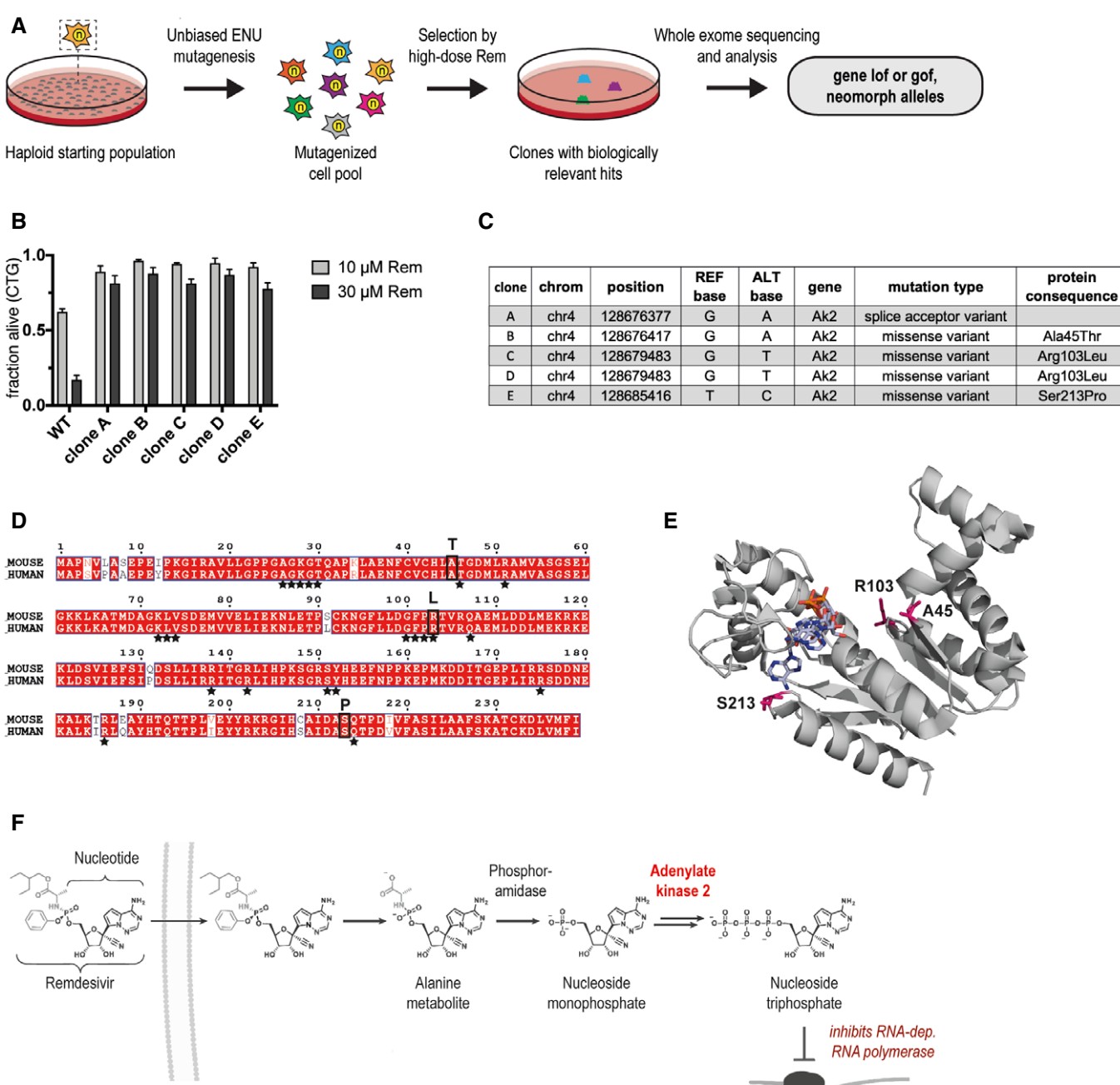

**Figure 2. Forward mutagenesis approach identifies remdesivir genetic interactions.**

A  Schematic of the chemical mutagenesis approach.

B  Cell viability of isolated cell clones and WT AN3–12 cells following 72 h remdesivir treatment with the indicated doses. Mean ± SEM of 2–4 biological replicates is displayed.

C  Ak2 mutations identified in the resistant clones analyzed in (B).

D  Protein sequence alignment of mouse and human AK2. Black stars indicate residues involved in nucleotide binding. Black boxes mark sites mutated in the identified cell clones with alternative residues indicated above.

E  Structure of human AK2 in complex with bis(adenosine)-5′-tetraphosphate (pdb:2c9y). Ak2 mutations identified in the remdesivir resistance screen are highlighted in magenta.

F  Schematic of remdesivir cellular uptake and intracellular activation. Modified from Eastman *et al* (2020).

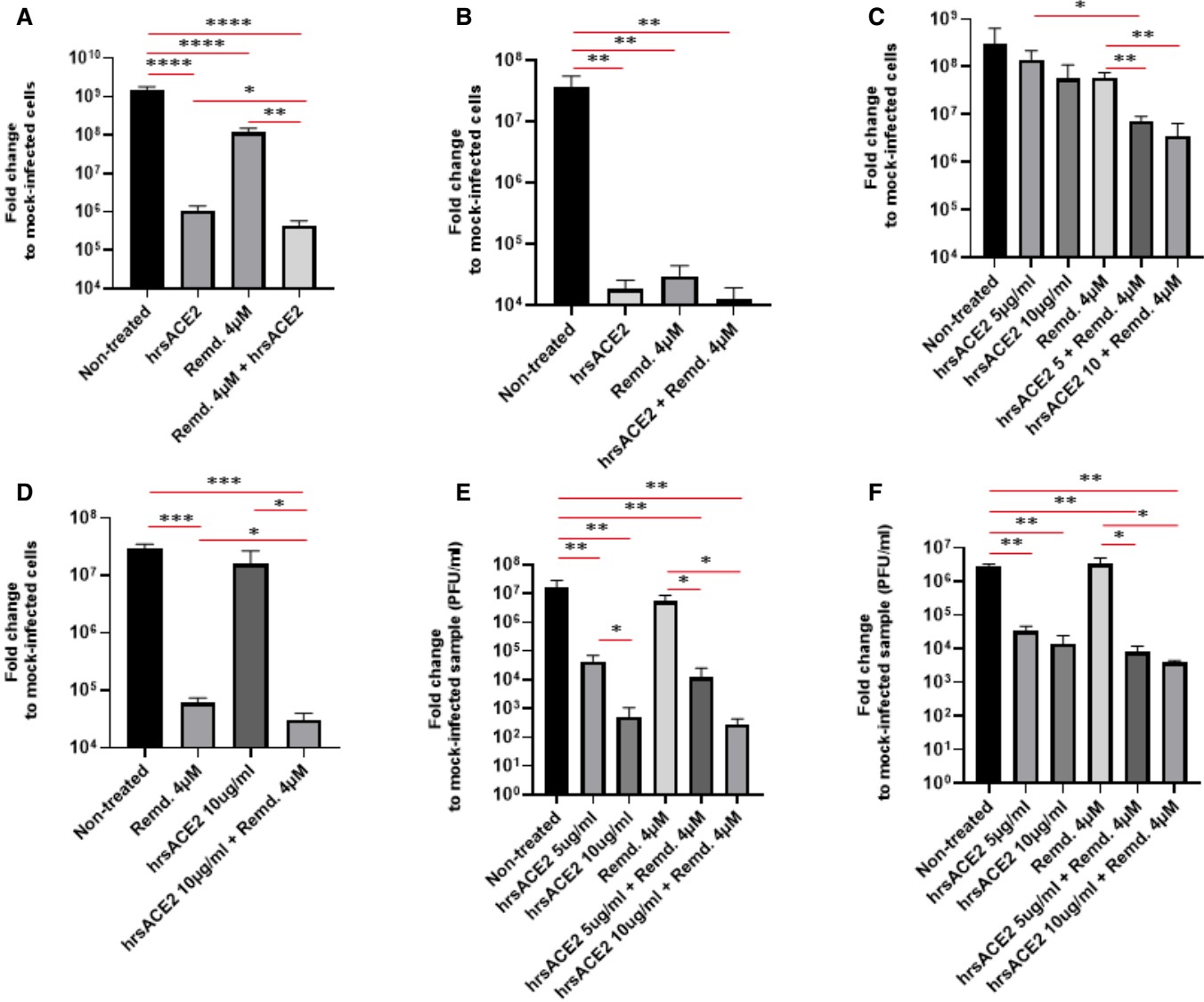

**Figure 3. Combined effect of remdesivir and hrsACE2 in blocking entry and replication of SARS-CoV-2 infections.**

A   Treatment of SARS-CoV-2 ($10^6$ PFU) infected Vero-E6 cells with human recombinant soluble ACE 2 (hrsACE2) (200 μg/ml) and remdesivir (Remd. 4 μM). Viral RNA level was determined at 15 h after virus inoculation.

B   Treatment of SARS-CoV-2 ($10^6$ PFU) infected human kidney organoids with hrsACE2 (200 μg/ml) and/or remdesivir (Remd. 4 μM). Viral RNA was determined by qRT–PCR 72 h after the inoculation of $10^6$ PFU of SARS-CoV-2.

C   Treatment of SARS-CoV-2 ($10^6$ PFU) infected Vero-E6 cells with clinical doses of hrsACE2 (5 and 10 μg/ml) and remdesivir (Remd. 4 μM).

D   Treatment of SARS-CoV-2 ($10^6$ PFU) infected kidney organoids with hrsACE2 (10 μg/ml) and remdesivir (4 μM).

E, F   Progeny virus released from untreated Vero-E6 cells or Vero-E6 cells treated with clinical doses of hrsACE2 (5 and 10 μg/ml) and remdesivir (Remd. 4 μM). Progeny was determined (E) 15 h and (F) 48 h post-infection (hpi).

Data information: Error bars show mean ± SD from biological triplicate. $n = 3$, *$P < 0.05$; **$P < 0.01$; ***$P < 0.001$; ****$P < 0.0001$; one-way ANOVA followed by Students $t$-test between internal groups. $P$-values are listed in Appendix Table S1.

Source data are available online for this figure.

## Material and Methods

### Virus

SARS-CoV-2 was isolated on Vero-E6 cells, from a nasopharyngeal sample of a patient in Sweden (Monteil *et al*, 2020). Virus titers were determined using a plaque assay as previously described (Becker *et al*, 2008) with fixation of cells 72 h post-infection. The

SARS-CoV-2 isolate was sequenced by Next-Generation Sequencing (Genbank accession number MT093571).

### Preparation of soluble recombinant human

Clinical-grade human recombinant soluble ACE2 (hrsACE2, APN01, amino acids 1–740) was produced by Polymun Scientific (contract manufacturer) from CHO cells according to Good Manufacturing

Practice guidelines and formulated as a physiologic aqueous solution (Haschke *et al*, 2013; Monteil *et al*, 2020).

### Liver and kidney cytotoxicity assays

To determine whether hrsACE2 or remdesivir at effective anti-viral doses are toxic to liver and kidney cells, we treated primary human liver spheroids (Bell *et al*, 2016; Stebbing *et al*, 2020) and human stem cell-derived kidney organoids (Garreta *et al*, 2019) with several concentration of hrsACE2 (50–800 μg/ml) or remdesivir (4–80 μM) in triplicate for 24 h. Three days post-treatment (for kidney organoids) and 15 h post-treatment (for liver spheroids) cytotoxicity (CC50) was determined using the CellTiter-Glo® Luminescent cell viability assay (Promega) following manufacturer's protocol using 50 μl of CellTiter-Glo® Reagent per well. Cytotoxicity of remdesivir in AN3–12 mouse embryonic stem cells was also assessed using the CellTiter-Glo® viability assay after 72 h incubation with the indicated remdesivir doses.

### Treatments of Vero E6 cells with hrsACE2 and remdesivir

Vero E6 cells were seeded in 48-well plates ($5.10^4$ cells per well) (Sarstedt) in DMEM containing 10% FBS. Twenty four hours post-seeding, dilution of remdesivir was prepared in DMEM 5% FBS in a final volume of 100 μl per well. Cells were treated with remdesivir or mock-treated for one hour. During this incubation time, hrsACE2 was mixed with the virus (1:1) in a final volume of 100 μl per well in DMEM (5% FBS) at 37°C for 30 min; then, remdesivir was added (or not) to the mixes before infection. Vero E6 were infected either with mixes containing hrsACE2/SARS-CoV-2, remdesivir/SARS-CoV-2, or hrsACE2/remdesivir/SARS-CoV-2 for 1 h. After 1 h, cells were washed three times with PBS and 200 μl of DMEM 5%FBS containing remdesivir 4 μM, hrsACE2 5 μg/ml or 10 μg/ml or combinations were added to the cells for 15 and 48 h. Fifteen hours post-infection, supernatants were removed and saved for titration, and cells were washed three times with PBS and then lysed using Trizol™ (Thermofisher) before analysis by qRT–PCR for viral RNA detection. Forty eight hours post-infection, supernatants were removed and saved for titration. Infectious progeny virus in supernatants was titered using a plaque assay with fixation of cells 72 h post-infection.

### Treatments of kidney organoids with hrsACE2 and remdesivir

The kidney organoid model for SARS-CoV-2 infection has been described recently (Monteil *et al*, 2020). Dilution of remdesivir was prepared in DMEM 5% FBS in a final volume of 100 μl per well. Kidneys were treated with remdesivir or mock-treated for one hour. During this incubation time, hrsACE2 (10 or 200 μg/ml) was mixed with 10^6 PFU of virus (1:1) in a final volume of 100 μl per well in Advanced RPMI medium (Thermo Fisher) at 37°C for 30 min; then, remdesivir was added or not to mixes before infection. Kidney supernatants were then removed, and kidneys were infected either with mixes containing hrsACE2/SARS-CoV-2, remdesivir/SARS-CoV-2, or hrsACE2/remdesivir/SARS-CoV-2 for 3 days. Three days post-infection, supernatants were removed, and kidneys were washed three times with PBS

and then lysed using Trizol™ (Thermofisher) before analysis by qRT–PCR for viral RNA detection.

### Mutagenesis screen, exome sequencing, and analysis

The screening procedure and the data analysis were extensively described previously (Horn *et al*, 2018). In brief, AN3–12 mouse embryonic haploid stem cells were cultured in DMEM high glucose (Sigma-Aldrich) supplemented with glutamine, fetal bovine serum (15%), streptomycin, penicillin, non-essential amino acids, sodium pyruvate, β-mercaptoethanol, and LIF. Cells were mutagenized with 0.1 mg/ml Ethylnitrosourea for two hours at room temperature 24 h prior to selection with 50 μM remdesivir. Two weeks later, resistant clones were isolated and subjected to remdesivir cytotoxicity assays and gDNA extraction using the Gentra Puregene Tissue Kit (Qiagen). Paired end, 150 bp whole exome sequencing was performed on an Illumina Novaseq 6000 instrument after precapture-barcoding and exome capture with the Agilent SureSelect Mouse All Exon kit. For data analysis, raw reads were aligned to the reference genome mm9. Variants were identified and annotated using GATK (v.3.4.46) and snpEff (v.4.2). Remdesivir resistance causing alterations were identified by allelism only considering variants with moderate or high effect on protein and a read coverage > 20.

### qRT–PCR

Samples were extracted using Direct-zol RNA MiniPrep kit (Zymo Research). qRT–PCR was performed using E-gene SARS-CoV-2 primers/probe following guidelines by the World Health Organization (https://www.who.int/docs/default-source/coronaviruse/wuhan-virus-assay-v1991527e5122341d99287a1b17c111902.pdf).

Forward primer: 5′-ACAGGTACGTTAATAGTTAATAGCGT-3'
Reverse primer: 5′-ATATTGCAGCAGTACGCACACA-3'
Probe: FAM-ACACTAGCCATCCTTACTGCGCTTCG-MGB
RNase P was used as an endogenous gene control to normalize the levels of intracellular viral RNA.
Forward primer: AGATTTGGACCTGCGAGCG
Reverse primer GAGCGGCTGTCTCCACAAGT
probe: FAM-TTCTGACCTGAAGGCTCTGCGCG-MGB

### Statistics

Statistical analyses were conducted using GraphPad Prism 8 (GraphPad), and significance was determined by one-way ANOVA followed by Student's *t*-test for internal groups. Error bars show mean ± SD from biological triplicate.

## Data availability

All source data of this study are available online. Other data that support the findings of this study are available from the corresponding authors upon request.

**Expanded View** for this article is available online.

## The paper explained

**Problem**
To date, there are no effective drugs for COVID-19. Only remdesivir has received authorization for COVID-19 and has been shown to improve outcomes but not decrease mortality.

**Results**
We have demonstrated an additive effect of combination therapy using remdesivir with recombinant soluble ACE2. This combination treatment markedly improved their therapeutic windows against SARS-CoV-2.

**Impact**
Our data lay the groundwork for the study of combinatorial regimens in future COVID-19 clinical trials and lead to better therapy in future.

## Acknowledgements

We thank all members of our laboratories for critical input and suggestions. This project has received funding from the Innovative Medicines Initiative 2 Joint Undertaking (JU) under grant agreement no. 101005026. The JU receives support from the European Union's Horizon 2020 research and innovation program and EFPIA. J.M.P. is supported by the Canada 150 Research Chair program, the von Zastrow foundation and COVID-19 grants from CIHR and the Austrian WWTF. This work was partially supported by the CIHR grants 440347, FDN143285, and OV3-170344. This work has received funding from the European Research Council (ERC) under the European Union's Horizon 2020 research and innovation program (StG-2014-640525_REGMAMKID to N.M.). A.M supported by Swedish research council (2018-05766). NM is also supported by the Instituto de Salud Carlos III (ACE2ORG), the Spanish Ministry of Economy and Competitiveness/FEDER (SAF2017-89782-R), the Generalitat de Catalunya and CERCA Programme (2017 SGR 1306), and Asociación Española contra el Cáncer (LABAE16006). C.H.P. is supported by Marie Skłodowska-Curie Individual Fellowships (IF) grant agreement no. 796590. N.M and C.H.P are supported by the EFSD/Boehringer Ingelheim European Research Programme in Microvascular Complications of Diabetes.

## Author contributions

VM performed all of the experiments involving SARS-CoV-2, including isolation, and helped with manuscript writing/editing. MD performed mutagenesis screening/sequence analysis. VML and SY conducted work with the liver spheroid system. VML helped with manuscript editing. HK performed all the qRT–PCR for virus involved experiment. GW, HZ, and ASS developed and produced clinical-grade hrsACE2. CHP developed derived kidney organoids. JMP, NM, MH, and AM designed the project and wrote the manuscript.

## Conflict of interest

JMP declares a conflict of interest as a founder and shareholder of Apeiron Biologics. GW is an employee of Apeiron Biologics. AS is a consultant to Apeiron Biologics. VML is co-founder, CEO, and shareholder of HepaPredict AB and discloses consultancy work for Enginzyme AB. MH is co-founder, CEO, and shareholder of Acus Laboratories GmbH and CSO of JLP Health GmbH. MD is employee of Acus Laboratories GmbH and JLP Health GmbH. All other authors declare no competing interests.

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
