## [Review Process File · EMBO Molecular Medicine]

Human soluble ACE2 improves the effect of Remdesivir in SARS-CoV2 infection

Vanessa Monteil, Matheus Dyczynski, Volker Lauschke, Hyesoo Kwon, Gerald Wirnsberger, Sonia Youhanna, Haibo Zhang, Arthur Slutsky, Carmen Hurtado del Pozo, Moritz Horn, Nuria Montserrat, Josef Penninger, and Ali Mirazimi

DOI: [10.15252/emmm.202013426](https://doi.org/10.15252/emmm.202013426)

Corresponding authors: Ali Mirazimi (ali.mirazimi@ki.se) , Josef Penninger (josef.penninger@imba.oeaw.ac.at)

Review Timeline:

Submission Date:	9th Sep 20
Editorial Decision:	5th Oct 20
Revision Received:	29th Oct 20
Editorial Decision:	5th Nov 20
Revision Received:	6th Nov 20
Accepted:	10th Nov 20

Editor: Zeljko Durdevic

Transaction Report:

5th Oct 2020

Dear Prof. Mirazimi,

Thank you for the submission of your manuscript to EMBO Molecular Medicine. We have now received feedback from the two reviewers who agreed to evaluate your manuscript. As you will see from the reports below, while the referee #2 is overall supporting publication of your work, referee #1 is more critical and raises number of concerns that should be addressed in a major revision of the current manuscript. Focus of the revision should be on addressing the criticism on the drug concentrations used, and on the models to test the drugs' toxicity. The experiments suggested by the referee #1 concerning the combination of hACE2 and remdesivir with neutralizing antibodies are not required for further consideration of the manuscript in our journal.

Addressing the reviewers' concerns in full, experimentally or in writing, will be necessary for further considering the manuscript in our journal, and acceptance of the manuscript might entail a second round of review. EMBO Molecular Medicine encourages a single round of revision only and therefore, acceptance or rejection of the manuscript will depend on the completeness of your responses included in the next, final version of the manuscript. For this reason, and to save you from any frustrations in the end, I would strongly advise against returning an incomplete revision.

Pending the appropriate revision, we think that rather than a "Correspondence" your work is more suitable for publication as a "Report" article. Therefore, I would like to ask you to do the following adjustments in the revised manuscript: 1) split the data from the Figure 1 into 2 figures; 2) add Material and Method section to the main text (current text is suitable as combined Results and Discussion section); 3) format the reference citation in the text and reference list according to the EMBO Molecular Medicine guidelines; 4) supplemental information should be formatted as an "Appendix" with a short table of contents. Please check our "Author Guidelines" for more information:

<https://www.embopress.org/page/journal/17574684/authorguide#reportsarticleguide>

<https://www.embopress.org/page/journal/17574684/authorguide#referencesformat>

<https://www.embopress.org/page/journal/17574684/authorguide#expandedview>

We would welcome the submission of a revised version within three months for further consideration. However, we realize that the current situation is exceptional on the account of the COVID-19/SARS-CoV-2 pandemic. Please let us know if you require longer to complete the revision.

***** Reviewer's comments *****

Referee #1 (Novelty/Model system Comments for Author):

Models used for this study are in vitro cell culture models. Vero cells for testing of therapeutic activity, kidney organoids and liver spheroids for toxicity testing.

Data from both models are difficult to translate into clinical reality.

Several substances were shown to be active in SARS-CoV-2 vero cell infection assays and failed in clinical trials. The best example for this is hydroxychloroquine.

Thus I would be cautious in using these in vitro data as the base (our groundwork as the authors state) for future clinical trials with the proposed combination.

With regard to toxicity, I am not sure whether the organoid data correlate with clinical and other in vivo data. Especially for remdesivir, extensive toxicity data from diverse models have already been published. The substance is approved for COVID-19 treatment in humans. What is the advantage of using organoids/spheroids at this time-point. How do data correlate with real life data from the clinical or animal trials?

https://www.ema.europa.eu/en/documents/other/summary-compassionate-use-remdesivir-gilead_en.pdf

In addition, I am not sure whether organoids/spheroids are the proper model to test for toxicity of a human recombinant protein (ACE2). This protein has also been tested in clinical trials and toxicity data must be available. See citations and comments below.

It is somewhat expected that a recombinant human protein fails to induce major toxicity in organoids when using reasonable concentrations.

There is not a single citation explaining or justifying the use of these cell systems for the respective research question.

Referee #1 (Remarks for Author):

In the correspondence paper entitled "Human soluble ACE2 improves the effect of Remdesivir in SARS-CoV2 infection", Monteil et al show in vitro efficacy data of combination therapy with remdesivir and recombinant soluble ACE2 in SARS-CoV-2 infection models.

In addition, in vitro toxicity data for the two drugs are provided by using liver spheroids and kidney organoids. Comments regarding relevance of these toxicity data are provided above (4. Adequacy of model system).

With regard to combination therapy the data show additive (or synergistic? - both are mentioned) effects for the two drugs when tested in vero cells (and kidney organoids) infected with SARS-CoV-2 leading to reduced detection of viral RNA and reduced viral replication in plaque assays.

My main issue with this study is that the drug concentrations used are somewhat random and justified by toxicity data derived from models with unknown relevance.

Wang et al. recently showed IC90 data for remdesivir of 1,76 uM. Monteil et al use 4 uM which is then called "low dose" remdesivir - this is not clear to me.

How do the 4 uM correlate with expected plasma levels upon human exposure? These data are available.

In addition, it is somewhat expected to see additive effects when using a combination of drugs that A) interfere with viral replication and B) block viral entry into cells.

Nevertheless, the in vitro finding may be of relevance for clinical trials using the two approaches, however, in my view the data are not sufficient to predict the outcome of these trials.

There is also a relevant set of published data for use of human ACE2. Even clinical trials with this protein have been published.

1) these data should be cited in the main text (not in the methods section)

<https://link.springer.com/article/10.1007%2Fs40262-013-0072-7>

<https://pubmed.ncbi.nlm.nih.gov/28877748/>

In fact it is surprising that these trials were not discussed in more detail by Monteil et al.

2) How do findings from these studies informed on hACE2 concentrations used in this in vitro study. In a first experiment 200ug/ml was used. This seems to be extremely high!

On the other side, are the lower concentrations used (5 and 10 ug) realistic concentrations when compared to the available PK data (Haschke et al.) and the expected plasma levels in the currently ongoing clinical trial NCT04335136?

As mentioned above, using hACE2 is an interesting approach for targeting SARS-CoV-2. However, there are several similar approaches using neutralizing antibodies which also block viral entry into the SARS-CoV-2 target cell. Multiple clinical trials are ongoing.

At first sight, pharmacokinetic attributes of these antibodies seem to be superior to hACE2. In vitro IC50, very long half life in humans.

It would be interesting to compare efficacy of hACE2 and remdesivir combinations to combinations with neutralizing antibodies.

hACE2 plus NAB - also additive?

Remdesivir plus NAB - also additive?

The manuscript would be strengthened by providing these data which would highlight even more treatment approaches for COVID-19.

Referee #2 (Remarks for Author):

This correspondence by Monteil et al. demonstrates the efficacy of combining lower doses of Remdesivir and soluble recombinant ACE2 in reducing SARS-CoV-2 viral load. They used in vitro cell culture and organoid models for the current set of experiments. Overall, results are encouraging and provide a groundwork for future clinical trials that might have better therapeutic outcomes.

I have the following minor comments:-

1. Abstract: The write-up needs to be improved. There are few grammatical errors, e.g. ACE markedly improve; should be "improves".

2. Abstract:their effectiveness.....; may be phrased differently to make it clearer.

3. Figures: The labeling for Y-axis "Fold-change to mock-infected cells" should be better defined. In some figures, Y-axis starts from 10^6 , while in others it starts with 10^5 , 10^4 , 10^3 , or 10^0 , which should be consistently presented.

Point by point reply to referee 1

Models used for this study are *in vitro* cell culture models. Vero cells for testing of therapeutic activity, kidney organoids and liver spheroids for toxicity testing. Data from both models are difficult to translate into clinical reality.

We thank the referee for the insightful review and the very important points raised. We agree with the reviewer that the choice of model system is essential to maximize the translatability of results. We selected liver spheroids based on our previous results in which we showed that this model system accurately recapitulates the molecular phenotypes of human liver on a proteomics, transcriptomic and metabolomic level for multiple weeks in culture (Bell et al. *Drug Metabolism and Disposition* **2017**, *45*, 419–429; Bell et al. *Sci. Rep.* **2016**, *6*, 25187; Vorrink et al. *FASEB J.* **2017**, *31*, 2696–2708). Moreover, as reported previously, this model outperforms other hepatic cell models and culture paradigms in multi-center trials (Bell et al. *Toxicol. Sci.* **2018**, *162*, 655–666; Bell et al. *Drug Metabolism and Disposition* **2017**, *45*, 419–429.). Further, in a large toxicity screen using 123 hepatotoxic drugs and non-toxic control drugs we found that the spheroid system was the most predictive model for drug-induced liver injury (Vorrink et al. *Toxicol. Sci.* **2018**, *163*, 655–665). These references has been cited in the revised MS. We used Kidney organoids and Vero-E6 as these are the model system we have used to investigate the antiviral activity of ACE-2 and remdesivir. In the revised manuscript, we have now included the reasons why we used these particular models (Page 5 line 136 to page 6 line 146). We do believe that the choice for liver and kidney models is relevant because these two organs have been reported to be affected by remdesivir toxicity in clinical trials (please see reply below for detailed discussion).

Several substances were shown to be active in SARS-CoV-2 vero cell infection assays and failed in clinical trials. The best example for this is hydroxychloroquine. Thus I would be cautious in using these *in vitro* data as the base (our groundwork as the authors state) for future clinical trials with the proposed combination.

We fully agree with the reviewer that no model can guarantee translatability. We selected Vero E6 cells as a commonly used infection model, which is backed up by robust and large data sets from multiple laboratories. To have such a vigorous model combined with our extensive experience using these cells is - in our opinion - major benefit, particularly for the study of novel and/or only partially understood viruses, such as SARS-CoV-2. Importantly, we validated our findings in 3D human kidney organoids. Of note, we were actually the first to introduce the use of organoids for SARS-CoV2 infections (Monteil et al. *Cell* 2020), i.e. we therefore have accrued extensive know-how to use these kidney organoids for SARS-CoV2 infection studies. In the revised manuscript, we make sure to further discuss the limitations of our study and to make clear to the reader that results from Vero E6 cells are only one (yet important) piece of data to initiate clinical studies

using this combination therapy (page 9, line 242). That is one reason why we added kidney organoids to have a second model system. Importantly, remdesivir has now been approved for clinical use in the USA and hrsACE2 is in advanced phase 2b stage clinical trials for severe COVID-19 patients, making this combination clinically feasible.

With regard to toxicity, I am not sure whether the organoid data correlate with clinical and other in vivo data. Especially for remdesivir, extensive toxicity data from diverse models have already been published. The substance is approved for COVID-19 treatment in humans. What is the advantage of using organoids/spheroids at this time-point. How do data correlate with real life data from the clinical or animal trials?

https://www.ema.europa.eu/en/documents/other/summary-compassionate-use-remdesivir-gilead_en.pdf

In healthy volunteers, C_{max} of remdesivir was found to be around 7.3µM after a 225mg dose (Humeniuk et al. *Clinical and Translational Science* **2020**, *13*, 896–906), which is in line with loading doses and pharmacokinetics (PK) in critically ill COVID-19 patients (Tempestilli et al. *J. Antimicrob. Chemother.* **2020**, *75*, 2977–2980.). Our data in both liver spheroids (TC₅₀=6.8µM) and kidney organoids (TC₅₀=10.5µM), indicates that toxicity is detected at concentrations that correspond to the C_{max} in patients, indicating that our in vitro data are relevant real-life patient treatment. As the reviewer correctly points out, toxicity of remdesivir is well described and we are citing two key references in our revised manuscript (Grein et al. *N Engl J Med* **2020**, *382*, 2327–2336; Mulangu et al. *N Engl J Med* **2019**, *381*, 2293–2303.). Therapeutic drug monitoring data that would allow to associate pharmacokinetic parameters with safety events is to the best of our knowledge not yet available. We thus assume that remdesivir exposure in those individuals experiencing drug-related toxicity approximates the range of the PK studies.

In addition, I am not sure whether organoids/spheroids are the proper model to test for toxicity of a human recombinant protein (ACE2). This protein has also been tested in clinical trials and toxicity data must be available. See citations and comments below. It is somewhat expected that a recombinant human protein fails to induce major toxicity in organoids when using reasonable concentrations. There is not a single citation explaining or justifying the use of these cell systems for the respective research question.

Hepatic and renal toxicities are among the most common adverse events in drug development. We agree with the reviewer that a lack of toxicity “is somewhat expected”; however, to confirm these assumption we decided to include the test of hrsACE2 in these models. We fully agree that *in vitro* tests cannot be exhaustive and have to be complemented with additional safety evaluations, e.g. in animals to test for potential adverse events in other organ systems. We have now clarified these points in the revised manuscript (page 7 line 211) and added citations for the clinical trials on hrsACE2 (with an acceptable safety profile based on phase 1 and phase 2 clinical studies) (page 3, line

78 and page 8 line 207) and also remdesivir reporting kidney and liver toxicity (in creaser AST, ALT, and creatinine) (page 5 line 137). Of course, for hrsACE2 one has to wait the data from the currently ongoing phase 2b, double blinded trial for a final evaluation of its potential *in vivo* toxicity.

In the correspondence paper entitled "Human soluble ACE2 improves the effect of Remdesivir in SARS-CoV2 infection", Monteil et al show in vitro efficacy data of combination therapy with remdesivir and recombinant soluble ACE2 in SARS-CoV-2 infection models. In addition, in vitro toxicity data for the two drugs are provided by using liver spheroids and kidney organoids. Comments regarding relevance of these toxicity data are provided above (4. Adequacy of model system).

Please see our responses above.

With regard to combination therapy the data show additive (or synergistic? - both are mentioned) effects for the two drugs when tested in vero cells (and kidney organoids) infected with SARS-CoV-2 leading to reduced detection of viral RNA and reduced viral replication in plaque assays.

We thank the referee for pointing this out. We have edited the revised manuscript and now always write “additive” effects.

My main issue with this study is that the drug concentrations used are somewhat random and justified by toxicity data derived from models with unknown relevance. Wang et al. recently showed IC₉₀ data for remdesivir of 1,76 uM. Monteil et al use 4 uM which is then called "low dose" remdesivir - this is not clear to me.

We apologize that this was not been clearly described in initially submitted manuscript, also because of space constraints, the paper being submitted as a brief correspondence. This has now been rectified in the revised paper expanded to be a Full Report, following discussions with the editor. The concentrations used in our report is based on our established infection assay in Vero E6, where we found an IC₅₀ of 4,02µM for remdesivir using 10⁶ PFU, equivalent MOI 20, of our Swedish SARS-CoV-2 isolate (first reported in Monteil et al. Cell 2020). In the revise paper we highlight discrepancy in IC₅₀/IC₉₀ of remdisivir in different reports depending the system used to measure IC₅₀/IC₉₀. For instance, Wang et al showed a IC₉₀ of 1,76µM on Vero E6 cells using a MOI of 0.05 for 48h whereas Pizzorno et al. (2020) showed a IC₅₀ of 0,99µM on Vero E6 infected with a MOI of 0.01 for 48h. Other studies showed yet again different values (reviewed by Frediansyah A. et al, 2020). Moreover, the impact of the method used to measure IC₅₀/IC₉₀ has been reported by Jeon et al, 2020 were the IC₅₀ for Remdesivir

was shown to 11,41 μ M on Vero cells (not Vero E6) infected at a MOI of 0,0125 for 24h while the IC50 was 8,24 μ M infected at a MOI of 0,05 for 48h. This is also the reason that we made sure to redo all these experiment in our model and with our SARS-CoV-2 isolate to have controlled side-by-side comparisons. We have now included a Table with IC50 values obtained in our infection assay (please see new Table 1) and adjusted the text accordingly to clearly address this issue (page 5 line 124 and page 7 line 199).

How do the 4 μ M correlate with expected plasma levels upon human exposure? These data are available.

As discussed above, in healthy volunteers, Cmax of remdesivir was found to be around 7.3 μ M after a 225mg dose (Humeniuk et al. *Clinical and Translational Science* **2020**, *13*, 896–906), which is in line with loading doses and pharmacokinetics (PK) in critically ill COVID-19 patients (Tempestilli et al. *J. Antimicrob. Chemother.* **2020**, *75*, 2977–2980). However, we are certainly aware that must be careful when discussing the concentration of an anti-viral in the body, especially for COVID-19, as most of the infection happens in organs such as lungs, blood vessels, heart or kidneys. Therefore, measurement of plasma concentrations of a compound are most probably not a great indicator, acknowledging that this might be the only way to record the PK in humans. The key finding of our work is that combining remdesivir with soluble ACE2, targeting two different modalities of the SARS-CoV-2 life cycle, namely cell entry via its receptor ACE2 and intracellular viral RNA replication, significantly improved their respective therapeutic window against SARS-CoV-2 and reduced the doses to much lower and apparently also safer levels.

In addition, it is somewhat expected to see additive effects when using a combination of drugs that A) interfere with viral replication and B) block viral entry into cells. Nevertheless, the in vitro finding may be of relevance for clinical trials using the two approaches, however, in my view the data are not sufficient to predict the outcome of these trials.

We thank the referee for stating that our data are relevant. We of course acknowledge that this is not a new concept - our findings are reminiscent of the breakthrough in HIV therapeutics that was bought about by simultaneous targeting of multiple distinct pathways in the viral replication cycle (Gulick, Mellors et al., 1997, Hammer, Squires et al., 1997). However, as far as we are aware, our data provide such evidence for the first time for SARS-CoV-2 infections and should lay the groundwork for the study of combinatorial regimens in future COVID-19 clinical trials. The outcome of such a combined therapy can of course only be determined in carefully designed clinical studies. Such studies appear even more pressing in light of the recently reported data from the WHO Solidarity trial that “remdesivir has little or no effect on hospitalized COVID-19, as indicated by overall mortality, initiation of ventilation and duration of hospital stay” (<https://www.medrxiv.org/content/10.1101/2020.10.15.20209817v1>).

There is also a relevant set of published data for use of human ACE2. Even clinical trials with this protein have been published. 1) these data should be cited in the main text (not in the methods section) <https://link.springer.com/article/10.1007%2Fs40262-013-0072-7> <https://pubmed.ncbi.nlm.nih.gov/28877748/>. In fact it is surprising that these trials were not discussed in more detail.

We have now added and cite these studies in the revised MS (Page 3 line 78). We now add and cite our name patient case report using hrsACE2 in a severe COVID-19, which we very recently published in Lancet Respiratory Medicine (Zoufaly et al. 2020). This paper for the first time reports data and the clinical course in a severe COVID-19 patient treated with hrsACE2.

2) How do findings from these studies informed on hACE2 concentrations used in this in vitro study. In a first experiment 200ug/ml was used. This seems to be extremely high! On the other side, are the lower concentrations used (5 and 10 ug) realistic concentrations when compared to the available PK data (Haschke et al.) and the expected plasma levels in the currently ongoing clinical trial NCT04335136?

To address this comment, as the referee correctly indicates, we refer to Haschke et al. who have previously reported the pharmacokinetic of hrsACE2, the same molecule we use for our current study, in healthy volunteers in a phase 1 clinical trial (Haschke, Schuster et al., 2013). These PK indicate that a concentration of 5-10 µg/ml of hrsACE2 is reached in plasma between 2 and 8 hours post-administration by administrating 800 µg/kg. The hrsACE2 dosing schedule (400 mg/kg every 12h for 7 days i.v.) for the phase 2b clinical trial in severe COVID-19 patients was in fact adjusted by an expert panel based on the reported PK data of Haschke et al. The key for this dose was to find the highest feasible dose to reach serum levels that are sufficient for viral neutralisation and have also highly significant enzymatic activities in the RAS for a prolonged time period. In our first studies in Vero E6 we used 200µg/ml of hrsACE2, but we used a very high dose of the virus, namely M.O.I 20. At a M.O.I. of 0.02, 25µg/ml of hrsACE2 were sufficient to effectively reduce the viral load. Thus, our data are also critical to guide the dosing of future hrsACE2 trial.

As mentioned above, using hACE2 is an interesting approach for targeting SARS-CoV-2. However, there are several similar approaches using neutralizing antibodies which also block viral entry into the SARS-CoV-2 target cell. Multiple clinical trials are ongoing. At first sight, pharmacokinetic attributes of these antibodies seem to be superior to hACE2. In vitro IC50, very long half-life in humans. It would be interesting to compare efficacy of hACE2 and remdesivir combinations to combinations with neutralizing antibodies. hACE2 plus NAB - also additive? Remdesivir plus NAB - also additive? The manuscript would be strengthened by

providing these data which would highlight even more treatment approaches for COVID-19.

We agree with the referee that this is an interesting question. However, to do that we would need to get access to these antibodies which are in clinical testing and therefore might be very difficult to get and it will take a long time to perform the same experiments side-by-side. We therefore hope that the referee agrees, and after discussions with the editor, that this is out of the scope of the current work. Of note, these antibodies are of course very interesting, however there will be a space for soluble ACE2, because it has an affinity to Spike that is similar to high affinity antibodies and in contrast to antibodies the virus will not be able “to mutate itself out of ACE2 binding”, otherwise the virus cannot fix to ACE2 and there will be no COVID-19.

Point by point reply to referee 2

This correspondence by Monteil et al. demonstrates the efficacy of combining lower doses of Remdesivir and soluble recombinant ACE2 in reducing SARS-CoV-2 viral load. They used in vitro cell culture and organoid models for the current set of experiments. Overall, results are encouraging and provide a groundwork for future clinical trials that might have better therapeutic outcomes.

We thank the referee for the insightful review and encouraging comments.

I have the following minor comments:-

1. Abstract: The write-up needs to be improved. There are few grammatical errors, e.g. ACE markedly improve; should be "improves".

We apologize for these mistakes. We were also asked now to expand our study from a correspondence to a full report, including abstract, introduction and a results/discussion section.

2. Abstract:their effectiveness.....; may be phrased differently to make it clearer.

We agree with the referee. We have reworked and expanded the abstract following the referee's comments and in agreement with the Editor. We also reworded therapeutic "effectiveness" to "therapeutic window", hoping this makes it clearer.

3. Figures: The labeling for Y-axis "Fold-change to mock-infected cells" should be better defined. In some figures, Y-axis starts from 10^6 , while in others it starts with 10^5 , 10^4 , 10^3 , or 10^0 , which should be consistently presented.

We have improved the Y axes labels in the revised manuscript to be consistent. Thanks for this important comment.

5th Nov 2020

Dear Prof. Mirazimi,

Thank you for the submission of your revised manuscript to EMBO Molecular Medicine. I am pleased to inform you that we will be able to accept your manuscript pending the following final amendments:

Please implement all adjustments suggested by the referee #1. No additional experiments are required.

***** Reviewer's comments *****

Referee #1 (Comments on Novelty/Model System for Author):

Comments regarding model systems made during the first revision round have been addressed sufficiently.

Referee #1 (Remarks for Author):

Most points of critique have been convincingly addressed by the authors.

Regarding the last issue asking for use of neutralizing antibodies in the model system, claiming that these antibodies are not available is not fully correct. There are multiple S-protein binding neutralizing monoclonal ABs available commercially. I recommend to at least mention the possibility of combining remdesivir and neutralizing ABs in the discussion.

Regarding the many issues with remdesivir in vitro activity, test systems and pharmacokinetics there now exist several extensive review articles addressing this issue. It might be helpful for the reader to refer to these in this article:

Malin et al., clinical microbiology reviews, 2020

Simonis et al., EMBO molecular medicine, 2020

Referee #2 (Comments on Novelty/Model System for Author):

Kidney organoids and Vero-E6 were used in the current studies. Use of these model systems for antiviral studies have previously been reported. Authors have explained the choice of using models in the revised version.

Referee #2 (Remarks for Author):

Overall, the revised version is much improved and authors have addressed my comments. I have nothing to add. It would be of great value to translate the work to the clinic.

The authors performed the requested changes.

The authors performed the requested changes.

Corresponding Author Name: Ali Mirazimi

Journal Submitted to: EMBO Mol. Biol.

Manuscript Number: EMM-2020-13426